# Cat–Owner Relationship and Cat Behaviour: Effects of the COVID-19 Confinement and Implications for Feline Management

**DOI:** 10.3390/vetsci9070369

**Published:** 2022-07-18

**Authors:** Giacomo Riggio, Carmen Borrelli, Patrizia Piotti, Anna Grondona, Angelo Gazzano, Francesco P. Di Iacovo, Jaume Fatjó, Jonathan E. Bowen, Daniel Mota-Rojas, Federica Pirrone, Chiara Mariti

**Affiliations:** 1Department of Veterinary Sciences, University of Pisa, 56124 Pisa, Italy; giacomo.riggio@phd.unipi.it (G.R.); carmen.borrelli@phd.unipi.it (C.B.); a.grondona@studenti.unipi.it (A.G.); angelo.gazzano@unipi.it (A.G.); francesco.diiacovo@unipi.it (F.P.D.I.); 2Department of Veterinary Medicine and Animal Sciences, University of Milan, 26900 Lodi, Italy; patrizia.piotti1@unimi.it (P.P.); federica.pirrone@unimi.it (F.P.); 3Chair Affinity Foundation Animals and Health, Department of Psychiatry and Forensic Medicine, School of Medicine, Autonomous University of Barcelona, 08003 Barcelona, Spain; jaume.fatjo@uab.es (J.F.); jbowen@rvc.ac.uk (J.E.B.); 4Queen Mother Hospital for Small Animals, Royal Veterinary College, North Mymms, Hertfordshire AL9 7TA, UK; 5Neurophysiology, Behavior and Animal Welfare Assessment, DPAA, Universidad Autónoma Metropolitana (UAM), Unidad Xochimilco, Mexico City 04960, Mexico; dmota@correo.xoc.uam.mx

**Keywords:** cat welfare, cat–owner relationship, COVID-19, cat–owner interactions, lockdown

## Abstract

**Simple Summary:**

The aim of this study was to explore the effects of the Italian COVID-19 lockdown on the cat-owner relationship and on cat behaviour. In order to do so, we used a questionnaire to obtain information on the owner and cat’s demographics, living environment, cat behaviour, and possible changes in some aspects of the cat-owner relationship. The questionnaire was distributed online when the lockdown was still ongoing and was completed by 548 cat owners, mainly women. Our findings suggest that the greater amount of time that owners spent at home with their cats, as a consequence of the social restrictions, had a positive effect on the owner’s perception of their relationship with their cat (e.g., higher frequency of interaction, greater emotional closeness, reduced perceived costs of caring for the cat). Similarly, the lockdown seemed to have improved the welfare of many cats, as suggested by the positive changes in many of their behaviours, even for those cats with pre-existing behavioural problems. Overall, our findings suggest that the lockdown provided many cats with a more thriving physical and social environment. They also highlight the need to reconsider some cat management practices commonly implemented in non-exceptional circumstances.

**Abstract:**

The restrictive measures implemented to stem the spread of COVID-19 abruptly changed the lives of many cats and their owners. This study explored whether the lockdown in Italy affected the cat–owner relationship, as well as cat behaviour and welfare. A survey that included questions on owner and cat’s demographics, living environment, cat behaviour and a modified version of the Cat/Dog Relationship Scale (C/DORS) was distributed online during the lockdown and was completed by 548 cat owners, mainly women (81.6%). With regard to the C/DORS subscales, both emotional closeness and cat–owner interactions increased during confinement, as opposed to a reduction in perceived costs. The effect of the type of job, family role and owner’s age on the C/DORS scores suggests that the relationship improved for those owners that, due to the lockdown, increased the time spent with their cats. For 58.8% of respondents, their cat’s general behaviour did not change, but when changes occurred, they were mostly positive (20.4%). Attention-seeking and demanding behaviours were the most increased during lockdown (25.7%). Cats with pre-existing problematic behaviours tended to either remain stable or improve during confinement. The overall positive effects of lockdown-related environmental changes on a cat’s behaviour suggest that some aspects of commonly implemented cat management practices should be revised to improve cat welfare in normal circumstances.

## 1. Introduction

On 11 March 2020, the COVID-19 outbreak was declared a pandemic by the World Health Organization. On the same day, the Italian government announced the beginning of the official lockdown, making Italy the first Western country to implement this strategy at national level. Just like in many other countries, the Italian lockdown was characterised by very restrictive measures on the citizens’ movement, social life, working life and recreational activities, with the aim to reduce the spread of the virus.

Consequently, the lockdown had a very strong emotional, psychological, physical, social and financial impact on the lives of many people whose daily routine was profoundly altered almost overnight. Living with a pet during the COVID-19 lockdown may have either mitigated or amplified the magnitude of the effects of confinement on each of these aspects of human life.

Pet ownership has been commonly associated with several positive psychological and physical effects on people’s wellbeing [1,2,3]. One way in which pets exert their positive effects is by functioning as a source of emotional and social support for their owners [4]. In fact, they can alleviate the negative impact of stressful conditions on the owner’s affective state [5].

In the specific scenario of the COVID-19 pandemic, multiple studies found that pets functioned as stress buffers for their owners. For instance, taking dogs for a walk [6] or engaging in social activities (e.g., affiliative, play) with them [7] was emotionally beneficial for the majority of dog owners. As for cats, in Jezierski et al.’s study [8], 84.6% of cat owners reported to have benefitted from the presence of their cat, mostly because it helped to alleviate the mental tension generated by the pandemic. This stress-mitigating effect may have led to the observed overall improvement in owners’ perception of their relationship with their pets during COVID-19 [9,10]. Bowen et al. [9] administered a modified version of the Cat/Dog–Owner Relationship Scale (C/DORS) to Spanish dog and cat owners, finding a general increase in their emotional closeness to the pet and frequency of pet–owner interactions, as well as a decrease in the perceived costs of the relationship, during the lockdown. Another study by Kogan et al. [10] that was focused on cat owners found that the bond to their animals strengthened significantly during the pandemic, partly because of the greater amount of time spent together.

While the positive role of pets during the pandemic is undeniable, pet owners seem to have experienced unique hardships that may have affected the perceived costs and benefits of the relationship with their pets, including concerns over accessing veterinary care, the emerging or worsening of pet behaviour problems and their pets’ role in the transmission of the virus [11]. Disease transmission concerns may have had a greater impact on cat owners’ attitude towards their animals [8], since both scientific [12] and public media sources [13] had soon raised doubts over a potential role of cats in the transmission of the virus.

There is evidence that the social and environmental changes that accompanied the restrictive measures implemented during the COVID-19 lockdown may have also affected the pets’ quality of life [14,15] and, consequently, their behaviour [9,11,16,17,18]. In a qualitative study by Shoesmith et al. [19], interviewing UK respondents, 67.8% of the owners reported changes in their pet’s behaviour and welfare during the lockdown. Surprisingly, in the specific case of cats, when behavioural changes were observed, they were mostly positive [19]. For instance, in 35.9% of cases, UK respondents described their cats as more affectionate, 27.2% as more relaxed, 4.1% as more anxious and only 1.1% as more hostile towards family members [19]. In a different study, 21.7% of Spanish owners believed their cats to be more relaxed, 36.4% to be more attention seeking, 7.0% to be more nervous and 2.2% to be more irritable during lockdown [9]. According to Bowen et al. [9], fear of loud noises and elimination outside of the litterbox were the two behavioural problems that were the most frequently affected by the lockdown—9.4% and 8.7% of cases, respectively—although noise fear mainly improved, whereas house soiling mainly worsened.

Italy is the fourth European country for the number of cats living in people’s households (7.9 mln) following Russia, Germany and France [20]. Additionally, only a relatively low number of studies focused on feline behaviour during the lockdown. Therefore, we aimed to explore the effects of the Italian COVID-19 lockdown on the cat–owner relationship and on cat behaviour.

## 2. Materials and Methods

This study obtained ethical approval from the Bioethical Committee of the University of Pisa, Italy (review no. 0040390/2020) and the Royal Veterinary College, UK (RVC SSRERB: URN SR2020-0171).

### 2.1. Subjects and Recruitment

The online questionnaire was created and hosted on the Jotform^®^ platform and distributed through an electronic link on Facebook^®^, Menlo Park, CA, USA (http//www.facebook.com, accessed on 12 May 2020). Responses were collected in April 2020. Respondents had to (1) be either dog or cat owners, (2) be at least 18 years old, (3) reside in Italy and (4) provide their informed consent to participate. Respondents who owned both species had to choose one; then, in case they owned more than one animal of the same species, in order to avoid potential bias on the selection of the animal, respondents were asked to answer for the one whose name was first in alphabetical order.

The questionnaire was distributed to both cat and dog owners, and a total of 1858 complete questionnaires were collected. In this study, only the responses from cat owners were included for analysis (28.0%).

### 2.2. Measurement

A slightly modified version of Bowen et al.’s [9] questionnaire, which included four different sections, was used in this study. The questionnaire with all the items is reported in the Appendix A. The first section collected information about the respondents (e.g., age, gender, profession, education level, role within the family, family composition) and their house (e.g., type, size, presence of an outdoor space). The second section investigated the effects of the confinement on the respondents and their families (e.g., duration of confinement period, permission to go to work, emotional, financial and health impact, impact on Quality of Life (QoL)). The third section comprised a modified version of the Cat/Dog–Owner Relationship Scale (C/DORS) to assess whether and how the pet–owner relationship had been affected by the confinement. Specifically, the five-point Likert scale used ranged from 1, “Much less than before the confinement”, to 5, “Much more than before the confinement”, with 3 meaning that no change was perceived compared to before the confinement. Scores for the perceived costs scale were reversed. Three items were removed from the original scale as they would either not be applicable to the lockdown period (item 27 “How often do you take your cat to visit people?” and item 29 “How often do you take your cat in the car?”) or they were considered to be too disturbing for owners to answer during an already potentially distressing time (e.g., item 25 “How traumatic do you think it will be for you when your cat dies?”). Additional questions directly investigated whether the owners perceived the lockdown to have affected their relationship with their pet, the pet’s QoL and whether they became angry with their pet more frequently than usual during the lockdown. The fourth section focused on the impact of the confinement on the pet’s behavioural problems. According to the species for which the respondent chose to answer, species-specific behavioural problems were investigated (e.g., urine marking for cats, aggression towards family members, destructive behaviours, etc.). In addition to specific behavioural problems, more general changes in the pets’ behaviour during confinement were also investigated (e.g., whether the pet was more relaxed, more stressed, more frustrated, more demanding). Since the targeted owners lived in Italy, all questions were originally written in Italian.

### 2.3. Statistical Analysis

Statistical analysis was performed with the software R [21]. The packages ordinal [22], rcompanion [23] and emmeans [24] were used for the regressions.

After computing the subscale scores for the modified C/DORS (emotional bond, cat–owner interaction, perceived costs), we calculated Spearman rho correlations between the modified C/DORS domains’ scores on one side and the behaviour problems. Behaviour problems with variance below 0.01 were excluded. Therefore, the behaviour problems “aggression towards unfamiliar people”, “aggression towards unfamiliar cats”, “house soiling”, “noise fear/sensitivity”, “destructiveness”, “hiding/avoiding contact with people” and “urine marking” were not included in further analysis. Alpha level was corrected with Bonferroni corrections for 27 multiple comparisons (α = 0.001).

We then calculated a series of ordinal regressions for each domain of the modified C/DORS and the behaviour problems “aggression towards familiar people” and “aggression towards familiar cats”. Each regression was calculated with fixed factors, without interactions: (1) a model of owner’s demographics included the age and gender of the owner; (2) a model of owner’s employment included job type and education; (3) a model of housing included the type of home, outdoor access, size, satisfaction in the size and in the access to outdoor and (for the behaviour problems) access to outdoor for the cat; (4) a model of finance included the family income; (5) a model of lockdown duration included the lockdown duration and the respondent’s expectations regarding the lockdown duration; (6) a model of lockdown consequences included financial consequences, emotional aspects, physical health and changes in lifestyle; (7) a model of the household included the number of underage family members (grouped as <6 years, 6–12 years, 13–17 years), the number of adults (grouped as 18–64 years, >64 years), the role of the respondent in the family, number of dogs, number of cats and characteristics of the family members; (8) a model of the demographics of the cat included age and sex; (9) a model of the relationship changes included changes in the QoL of the cat during the lockdown, comfort of the cat during the lockdown and whether the owner felt angry with the cat.

For each model, we ran an ANOVA to compare the Akaike Information Criterion (AIC) with that of a null model that included only the slope to test for model fit. When the full model was a better fit, we calculated the explained variance (R_pseudo_^2^) using the Nagelkerke method and we ran a pair-wise post hoc analysis with Tukey correction for the categorical predictors.

## 3. Results

### 3.1. Demographics of the Owners and Their Cats

The questionnaire was completed by 548 cat owners, mostly females (81.6%). The most represented age group was 36–45 years old (30.7%). More than half of the respondents had either a degree or a post-graduate degree (60.0%). With regard to the composition of the household, 72.2% included one or two 18 to 64 year-old adults, and 45.7% had at least one child between 0 and 17 years old. Most of the respondents lived in apartments (61.5%) and, although 29.7% of them reported to live in a small house, 86.1% considered this to be big enough for all of the family members. The great majority of the houses had some sort of outdoor access (92.8%), which included a private garden/courtyard in 44.0% of cases. The overall family income was considered average for 73% of the respondents, high for 23% and low for the remaining 4%. At the time of their participation in the study, 71.7% of the respondents had been in lockdown for 4 or 5 weeks. However, some of them or their family members were still permitted to travel to their workplace. According to our results, for 50.6% of respondents, all of the family members were confined, for 35.8%, some of them were confined while others were able to go to work, and, for 13.7%, all members were permitted to go to work. Appendix A provides a complete summary of the owners’ demographics, as well as of the information about the living environment.

The number of household cats ranged from one to six (median = two), and, in 39.8% of cases, at least one dog was also present. Cats were mostly spayed females (48.9%) or castrated males (46.0%), while only 2.4% of them were intact males or 2.7% intact females. Their age ranged from less than 1 year to more than 16 years (median = 6 years). Almost half of the cats were allowed outside both before and during the confinement (44.7%), whereas more than a half of them were indoor only, in either condition (51.6%). A few owners changed the type of management during the lockdown by either starting to provide outdoor access to cats that were previously kept only indoors (2.5%), or by denying outdoor access to cats that were previously allowed outside (1.1%). Appendix A provides a complete summary of the cats’ demographic and behavioural information.

### 3.2. The Effect of Confinement on Cat-Owner Relationship

When directly asked how the confinement period affected their relationship with their cats, 55.1% of the respondents reported some degree of improvement, 43.7% reported no change and only 1.2% reported the relationship to have worsened.

The mean scores for each C/DORS subscale showed that the owners perceived an overall increase in the emotional bond (mean = 3.39, SD = 0.39) and the cat–owner interaction (mean = 3.59, SD = 0.44) dimensions, along with a decrease in the perceived costs of the relationship dimension (items in this subscale are reversed) (mean = 3.16, SD = 0.33).

Ordinal regression indicated that the variance in the cat–owner interaction dimension (Figure 1) could be explained by models built on household characteristics (6%) (Model Fit: Nagelkerke Rp2 = 0.06, *p* = 0.0005), owners’ education and occupation (5%) (Model Fit: Nagelkerke Rp2 = 0.05, *p* = 0.0005) and owners’ physical features (4%) (Model Fit: Nagelkerke Rp2 = 0.04, *p* = 0.002). For the first model, post hoc pairwise comparisons revealed that the respondents who indicated their role within the family to be that of sons or daughters were more likely to have increased their interactions with their cat during the lockdown compared to respondents who defined themselves as adult individuals without children. As for the second model, owners who worked as labourers were more likely to have increased interactions with their cat compared to those who worked with animals (β = 1.427, *p* = 0.043), those who worked as stay-at-home (β = 1.893, *p* = 0.007) and those who were retired (β = 1.990, *p* = 0.005). Retired owners were less likely to have increased their interactions with their cat compared to owners who worked as employees (β = −1.219, *p* = 0.027). As for the third model, owners in the 26–35 age range were more likely to have increased interactions with their cat compared to owners in the 56–65 age range (β = −0.784, *p* = 0.044).

The variance in the perceived emotional closeness dimension (Figure 2) was explained by models built on owners’ education and occupation (7.4%) (Model Fit: Nagelkerke Rp2 = 0.074, *p* = 0.000013), and owners’ physical features (3.4%) (Model Fit: Nagelkerke Rp2 = 0.034, *p* = 0.0075). As for the first model, owners who worked as employees (β = −1.108, *p* = 0.0001), those who worked as freelancers (β = −0.944, *p* = 0.003) and those who worked as labourers (β = 1.491, *p* = 0.031) were more likely to have improved their emotional bond with their cats compared to owners who worked with animals. With regard to the second model, female owners were more likely to have improved the bond with their cat compared to male owners (β = −0.734, *p* = 0.0007).

No significant effect was detected for any of the models tested on the perceived costs dimension of the C/DORS.

Participants were directly asked how their cat helped them during the lockdown compared with before. For 85.2% of them, their cat provided more support during the lockdown than before, whereas 10.8% did not perceive any change and only 4.0% perceived a reduction in the supportive role of their cat. Finally, 12.9% of the owners got less angry with their cat during the lockdown than before, 75.9% reported no difference and 11.1% got more angry with their cats.

### 3.3. The Effect of Confinement on Cat Behaviour and Welfare

When responding to how the lockdown affected their cat’s quality of life, more than half of the owners (56.2%) reported that it improved, 39.4% reported that it did not change and only 4% reported that it worsened.

Owners were also asked to report on possible changes in their cat’s general behaviour during the lockdown. The majority of them (58.8%) reported no changes. Only 4% of the owners reported at least one negative change, such as being more stressed, more nervous, more frustrated or more irritable, whereas 20.4% reported at least one positive change, such as being calmer or more relaxed. Finally, 25.7% reported their cat to be more demanding or attention-seeking.

Specific problematic behaviours were also investigated. Most of the cats (from a minimum of 90.9% for inappropriate elimination to a maximum of 98.5% for aggression towards other household cats) either never manifested the problem or their behaviour was not affected by the lockdown. For all behaviours, except fear of loud noises, the percentage of cats that improved during the lockdown was higher than the percentage of cats that worsened. A scatterplot graph of the Spearman’s correlations between the modified C/DORS domains’ scores on one side and the behaviour problems is reported in Appendix A. Ordinal regressions revealed that 8% of the variance for the variable “aggression towards family members” (Figure 3) was explained by the model including household-related predictive variables (Model Fit: Nagelkerke Rp2 = 0.08, *p* = 0.0005). A post hoc pairwise analysis revealed that respondents who lived in households in which some members were confined and some had permission to go out to work were more likely to report their cat’s aggressive behaviour towards family members to be worsened during the lockdown, compared to households in which all members had permission to go to work (β = −0.205, *p* = 0.042). Furthermore, 3% of the variance for this variable was explained by the cat demographics model (Model Fit: Nagelkerke Rp2 = 0.03, *p* = 0.0021), with spayed female cats being more likely to have this behaviour worsen during the lockdown compared to castrated males (β = 0.683, *p* = 0.003).

As for the variable “aggression towards other household cats” (Figure 4), 2.6% of the variance was explained by the model including house-related predictors (Model Fit: Nagelkerke Rp2 = 0.026, *p* = 0.032). Specifically, owners who perceived their house to be too small for all household members were more likely to report increased “aggression towards other household cats” during the lockdown, compared to owners who perceived their house to be big enough (β = 0.514, *p* = 0.041). No other significant effect of the models tested was found for any other problematic behaviour.

## 4. Discussion

The aim of this study was to investigate the possible effects of the COVID-19 confinement on the cat–owner relationship, as well as on cat behaviour and welfare. Our data were collected while the lockdown was still ongoing and the feeling of uncertainty for the future still heavily affected many people’s emotional and psychological well-being.

Our findings support the results from previous studies, indicating an overall positive effect of the COVID-19 confinement on the cat–owner relationship [8,9]. When directly asked how the lockdown affected the relationship with their cat, more than half of the respondents reported some degree of improvement, whereas almost half of them reported no change at all. Furthermore, the percentage of owners who felt that their cat provided more support during the lockdown compared with before reached approximately 85%, as opposed to a minority of owners who perceived a reduction in their cat’s support. The response pattern is similar to that reported by Bowen et al. [9] for the same question, although their respondents were both dog and cat owners. Similarly, Jezierski et al. [8], who specifically investigated cat owners, found that 75% of them indicated that their cat helped to alleviate their emotional tension during the lockdown.

There is a vast body of research about the positive effects of pets on people’s psychological wellbeing [2,25,26,27,28,29]. While these may be exerted through different mechanisms [30] that are not mutually exclusive, research often agrees on the pets’ ability to operate as a source of social support for their human companions [7,31,32,33]. According to a recent study by Janssens et al. [5], in stressful conditions, such as the one generated by the COVID-19 lockdown, the sole presence of a pet may buffer the negative effects of stress by helping people to retain their positive emotions. With specific regard to cats, Stammbach and Turner [34] found that their social support function increases along with the owners’ level of attachment to them. Although we did not specifically investigate their attachment to cats, the owners from the current study reported an overall increase in emotional closeness to their cats during confinement. As in Bowen et al.’s study [9], the increase in emotional closeness was accompanied by an increase in the frequency of cat–owner interactions and a reduction in the owner’s perceived costs of caring for the cat, confirming the overall positive effect of the lockdown on the owners’ perception of the relationship with their cat. When investigating possible effects of several demographic variables on the scores of each C/DORS dimension, we found that those categories of owners who had most likely increased the time spent at home, as a consequence of the social (e.g., young adults) and working (e.g., dependent workers, not “essential” workers) restrictions, were also more likely to report an increase in both interactions with and emotional closeness to their cats. For instance, respondents who indicated their role within the household to be that of a son or a daughter, as well as those who were aged between 26 and 35 years, were more likely to have increased the frequency of interactions with their cat during the lockdown compared to those who identified themselves as childless and those between 56 and 65 years old, respectively. Furthermore, those who worked as labourers or employees, whose working shifts and/or working location had likely changed during the lockdown, were also more likely to have increased the number of interactions with their cat compared to those who were already retired. On the contrary, owners who worked jobs that were less affected by the lockdown in terms of time spent out of home, either because they already worked inside the house (e.g., stay-at-home parents) or because they belonged to categories with special working permissions, i.e., who could work even during lockdown as their job was considered necessary (e.g., animal caretakers, veterinarians, etc.), were less likely to have increased the interactions with their cats. Employees, labourers and freelancers were also more likely to report an increased emotional closeness to their cats compared to owners who worked with animals. Therefore, these findings suggest that it is the increased amount of time that owners were able to spend with their cats during the lockdown that had an impact on their relationship. Similarly, Kogan et al. [10] reported that over 60% of cat owners felt that spending more time with their cat during the lockdown strengthened their bond.

We also found gender to influence the scores of the perceived emotional closeness dimension of the C/DORS, with female owners being more likely to report that the bond with their cat improved during the lockdown. According to previous research, women tend to score higher than men on scales about attachment to and social support from pets [35,36]. Furthermore, they tend to value and seek social support more than men during stressful situations [37,38].

As for the last dimension of the C/DORS, we found that owners perceived the costs of the relationship with their cat to be lower during confinement than before. It should be noted that several items in this dimension measure the extent to which the pet interferes with the owner’s freedom to perform other activities. Therefore, as suggested by Bowen et al. [7], who obtained very similar results, it is conceivable to argue that the social restrictions that the owners had to face during the lockdown were no longer attributed to the cats, but rather to the COVID-19-related regulations implemented by the government. Nonetheless, it is also possible that the C/DORS items do not address those peculiar hardships that were found to be associated with pet-ownership during confinement. Specifically, owners at this time have faced concerns over the access to veterinary care and pet supplies, the management of new behavioural issues (e.g., being more demanding and attention-seeking in the case of cats) or the fear of cross-species transmission of the virus [10,11], which may have altered the cost–benefit balance in ways that were not measured by the C/DORS items.

It should be considered that, when we use scales as tools to assess the quality of the pet–owner relationship, we are actually assessing the subjective perception of the human involved in that relationship, which does not necessarily reflect the perspective of the animal [39,40]. Moreover, within the specific framework of the social exchange theory [41], the perception of the costs of caring for the pet may be affected by the (mis)beliefs and (mis)conceptions that owners may have of that species. Contrary to dogs, indoor cats are often wrongly considered capable of easily tolerating the absence of their owners, or [42] thriving even in small and confined spaces [43], with possible serious consequences for their welfare [44,45,46]. For these reasons, cat owners may not perceive their pets as an impediment to their daily outdoor activities, and this, in turn, may affect their perception of the costs of caring for the cat. Being aware of the cat’s social needs may lead to higher perceived costs in normal circumstances and to an even greater decrease in perceived costs during confinement than that observed in this study, similar to what has been previously found for dog owners [9,43]. However, we did not assess the degree of owners’ awareness of cats’ ethological and relational needs. Therefore, we can only suggest that future studies using the social exchange paradigm to explain aspects of the pet–owner relationship, both in comparisons across species and comparisons between different environmental conditions, always take possible links between perceived costs, common management practices and welfare issues into account.

One aim of this study was also to assess whether the lockdown had an impact specifically on cat behaviour and welfare. In accordance with Bowen et al.’s [9] findings, more than half of the respondents reported that their cat’s quality of life improved during the lockdown. Moreover, when asked about general aspects of their cat’s behaviour, more than 40% of the owners reported at least one behavioural change during the lockdown, which was positive (e.g., more calm, more relaxed) in more than 20% of cases and negative (e.g., more irritable, more frustrated) only in 4% of cases. Interestingly, the most frequently reported behavioural change was the cat being more demanding and seeking more for the owner’s attention. We did not categorise these two behaviours as either positive or negative in terms of cat wellbeing, as their valence may depend on the motivation behind them and on the owner’s response. Although they may as well be early indicators of emotional disorders, as suggested by Bowen et al. [9], the overall positive changes in cat’s general behaviour seem to suggest otherwise. A possible explanation is that owners were simply more aware of this behaviour, as they spent more time at home with their cat. It is also possible that the increase in the owners’ availability—often the only social resource for a house cat—led to an actual increase in the cat’s motivation to seek social interaction. While there is evidence that some cats may not tolerate some forms of human interaction well, there is also evidence that most house cats do enjoy physical contact with their owners [47]. Furthermore, a previous study by Heidenberger [44] found that owners who spent more time interacting with their cats and over the entire day rather than only in the morning and/or in the evening also reported fewer behavioural problems. This supports our findings that a condition, such as the lockdown, that allowed owners to spend more time interacting with their cats and not just at very limited hours of the day had a simultaneous positive effect on cat behaviour. Furthermore, for many indoor cats, owners may represent the main source of stimulation, in otherwise physically and socially impoverished environments [48].

However, our results also suggest that owners may represent a source of stress for their cats in some circumstances [49]. In fact, when we investigated the effect of demographic variables on the evolution of multiple behavioural problems that were already manifest prior to the beginning of the lockdown, we found that aggression towards family members was more likely to worsen when at least some members of the household were confined and some had permission to go out to work rather than when all members had permission to leave the house for work. Similarly, aggression towards other household cats was more likely to have worsened if owners reported their house to be too small to allow for all family members to perform their activities separately. This definition represents a subjective measure of density rather than absolute space, and it is therefore related to a social rather than a physical aspect of the cat’s living environment. Overall, a possible interpretation is that, for those cats that already display signs of psychological or emotional discomfort in the form of aggressive behaviour, an increase in the amount of time spent with the owners and the impossibility of isolating themselves due to a high household density may have negative consequences for their welfare.

However, these findings need to be contextualised in light of an overall lack of effect of confinement on most cats with pre-existing behavioural problems. In fact, no change was reported in 71.2% to 95.8% of these cats, depending on the behavioural problem investigated. Furthermore, when an effect of confinement was detected, this was mainly positive for all behavioural problems, except for fear of loud noises, reflecting Bowen et al.’s [9] findings.

A limitation of this study is the relatively low number of respondents. We had an overall low number of cats whose behaviour changed during confinement, with a subsequent low variance. This may have affected the results of the regression analysis. Larger samples are therefore required to obtain more detailed information and draw more well-founded conclusions about the effect of social and environmental variables on cat behaviour during the COVID-19 lockdown.

In addition, our sample of respondents consisted mainly of female owners. Greater female involvement in human–animal relationship studies is not uncommon and cannot be avoided when participants are self-selected. In addition, self-selected participants may bias the sample towards owners that have a more positive perception of the relationship with their cat, as well as a more positive attitude towards them. Therefore, we should assume that our sample may not be representative of the Italian cat owner population.

## 5. Conclusions

The results from the current study support previous findings on the effect of the COVID-19 confinement on the cat–owner relationship, as investigated in other European countries. Overall, confinement seemed to have a positive effect on the owner’s perception of the relationship with their cat. Owners who increased the amount of time spent with their cats reported a higher frequency of interactions, as well as a stronger bond, compared with before the confinement. Furthermore, although many cats did not seem to be affected by the environmental and social changes that came with the lockdown, several others seemed to benefit from them. Owners often reported their cats to be more relaxed and calmer, although they were most commonly described as more attention-seeking and demanding. These findings may have implications for house cat management practices in exceptional, as well as in normal, circumstances, as they suggest that cats may benefit from conditions that entail a greater amount of time spent in the company of their owners.

## Figures and Tables

**Figure 1 vetsci-09-00369-f001:**
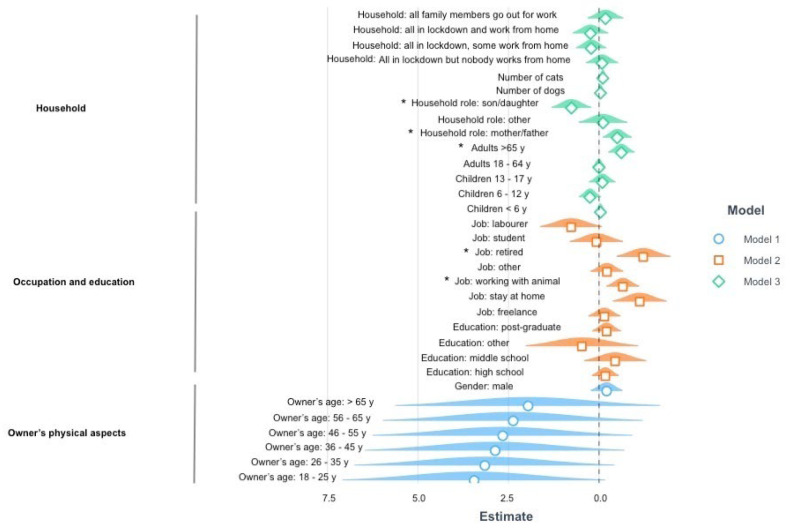
Results of the ordinal regressions on the domain cat–owner interactions. The graph represents the estimates (β) and standard errors (s.e.) for the significant models explaining the variance in cat–owner interactions during COVID-19 confinement. The dotted line represents the intercept, the geometrical figures represent the estimates and the coloured shadows represent the standard errors. When the response options were categorical, rather than binary or continuous, the reported parameters are compared against the reference parameter for each variable: some are in lockdown and some can go to work for the lockdown of the household, adults without children for the household role, employee for the job, degree for the education, female for the gender of the owner. Significant comparisons are indicated with an asterisk. Positive estimate values are on the left, whereas negative estimate values are on the right.

**Figure 2 vetsci-09-00369-f002:**
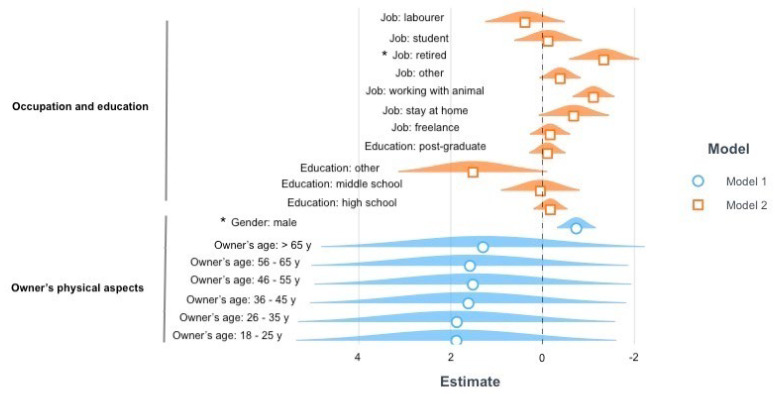
Results of the ordinal regressions on the domain perceived emotional closeness. The graph represents the estimates (β) and standard errors (s.e.) for the significant models explaining the variance in the perceived emotional closeness during COVID-19 confinement. The dotted line represents the intercept, the geometrical figures represent the estimates and the coloured shadows represent the standard errors. When the response options were categorical, rather than binary or continuous, the reported parameters are compared against the reference parameter for each variable: employee for the job, degree for the education, female for the gender of the owner. Significant comparisons are indicated with an asterisk. Positive estimate values are on the left, whereas negative estimate values are on the right.

**Figure 3 vetsci-09-00369-f003:**
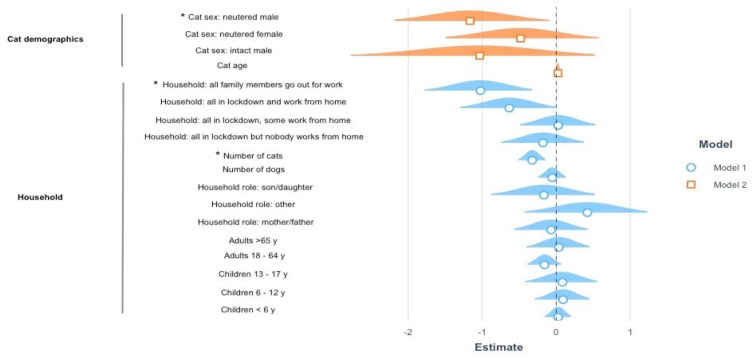
Results of the ordinal regressions on aggression towards familiar people. The graph represents the estimates (β) and standard errors (s.e.) for the significant models explaining the variance in the aggression towards familiar people during COVID-19 confinement. The dotted line represents the intercept, the geometrical figures represent the estimates and the coloured shadows represent the standard errors. When the response options were categorical, rather than binary or continuous, the reported parameters are compared against the reference parameter for each variable: intact female for the sex of the cat, some are in lockdown and some can go to work for the lockdown of the household, adults without children for the household role, employee for the job, degree for the education, female for the gender of the owner. Significant comparisons are indicated with an asterisk. Positive estimate values are on the right, whereas negative estimate values are on the left.

**Figure 4 vetsci-09-00369-f004:**
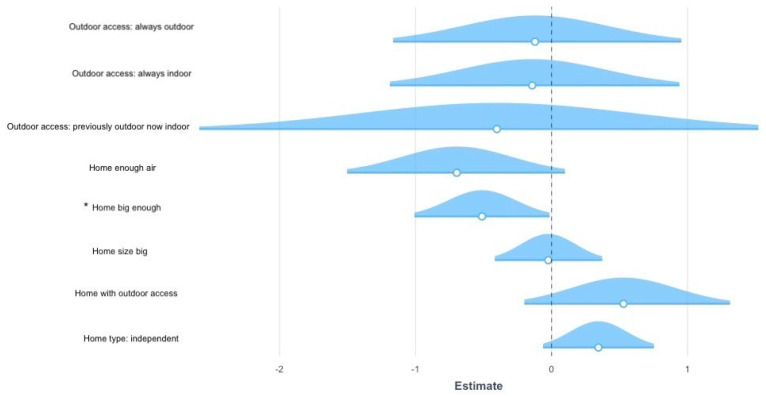
Results of the ordinal regression on aggression towards familiar cats. The graph represents the estimates (β) and standard errors (s.e.) for the significant models explaining the variance in the aggression towards familiar cats during COVID-19 confinement. The dotted line represents the intercept, the geometrical figures represent the estimates and the coloured shadows represent the standard errors. When the response options were categorical, rather than binary or continuous, the reported parameters are compared against the reference parameter for each variable: previously indoor and now outdoor for the outdoor access, flat for the home type. Significant comparisons are indicated with an asterisk. Positive estimate values are on the right, whereas negative estimate values are on the left.

## Data Availability

Data are available on request from the corresponding author.

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
