# Peer review of "Cat–Owner Relationship and Cat Behaviour: Effects of the COVID-19 Confinement and Implications for Feline Management"

_vetsci, 2022, doi:10.3390/vetsci9070369_

Round 1
Reviewer 1 Report
The present manuscript is laid out in a clear and precise manner. I found it an interesting read. The aim of this study was to investigate the possible effects of the COVID-19 confinement on the cat-owner relationship, as well as on cat behaviour and welfare. These would be very topical areas of interest to cat owners and the veterinary personal dealing with these clients. In fact it would have been interesting if the surveys had incorporated questions on how owners managed pets with "bad behaviour". As the authors used a slightly modified version of Bowen et al.’s questionnaire It would be helpful for the reader to be able to see the actual survey questions used and maybe this could be added as an addendum to the manuscript?
Regarding figure 4, the following parameters require a definition:
"enough air space",
"home big enough",
"home size big",
"living in an independent house"
In lines 289 -290 were respondees given guidelines as to what was meant by the parameters listed above? Or, was it a subjective interpretation made by the owners?
There were some grammatical and spelling mistakes:
Line 96:"This study obtained ethical approval from the Bioethical Committee.."
Line 115: Please give the full name for QOL as it is the first time it is being used in the text.
Line 145: correct spelling of "descructiveness"
Line 165: correct tense for "..run an ANOVA..."
Line 167: correct tense for "..run a pair-wise.."
Line 301: correct tense for "..our data were collected.."
Line 307: rephrase: "while little less than half.."
Line 309: rephrase: "compared to the previous time"
Line 482: correct spelling of "Journai"
Reference 35: please rewrite
Author Response
Responses to Reviewer I:
The present manuscript is laid out in a clear and precise manner. I found it an interesting read. The aim of this study was to investigate the possible effects of the COVID-19 confinement on the cat-owner relationship, as well as on cat behaviour and welfare. These would be very topical areas of interest to cat owners and the veterinary personal dealing with these clients. In fact it would have been interesting if the surveys had incorporated questions on how owners managed pets with "bad behaviour". As the authors used a slightly modified version of Bowen et al.’s questionnaire It would be helpful for the reader to be able to see the actual survey questions used and maybe this could be added as an addendum to the manuscript?
We thank the reviewer for their comments and suggestions. We added the complete version of the survey as supplementary material.
Regarding figure 4, the following parameters require a definition:
"enough air space",
"home big enough",
"home size big",
"living in an independent house"
In lines 289 -290 were respondees given guidelines as to what was meant by the parameters listed above? Or, was it a subjective interpretation made by the owners?
As for the first two parameters (enough air space, home big enough), only yes and no were given as possible answers. As for the size of the home, it could be either small, medium, big. As for the last parameter, they were asked what type of house they lived in, with different options as possible answers (e.g. apartment, independent house, etc.). In order to make both questions and response options clearer to the reader, we attached the actual questionnaire, as suggested by the reviewer.
There were some grammatical and spelling mistakes:
Line 96:"This study obtained ethical approval from the Bioethical Committee.."
Correction made
Line 115: Please give the full name for QOL as it is the first time it is being used in the text.
Full name provided along with acronym
Line 145: correct spelling of "descructiveness"
Correction made
Line 165: correct tense for "..run an ANOVA..."
Correction made
Line 167: correct tense for "..run a pair-wise.."
Correction made
Line 301: correct tense for "..our data were collected.."
We do not see the error here. Perhaps the reviewer may be more specific.
Line 307: rephrase: "while little less than half.."
Rephrased as “while almost half of them”
Line 309: rephrase: "compared to the previous time"
Rephrased as “compared with before”
Line 482: correct spelling of "Journai"
Correction made
Reference 35: please rewrite
Reference rewrote
Reviewer 2 Report
Dear authors,
I have found your work of interest. However I have a few comments.
Broad comments
1. L344 I think that would be of interest discussing directly why in professionals working with animals time spent with their own cats was not increased. Seems a valuable result of the study. The generalist term "special working permissions" seems harder to understand
2. In the regression graphics, I would suggest for clarity including some reference to interpret that increased values comparing with pre-covid values are on the left, and decreased on the right of the estimate 0.0 .
Specific comments
L165 Please describe the whole words of "AIC", is not so well known as ANOVA or other general terms (e.g)
L78 Please describe the first time, the use of QoL, you didn't mention the acronym.
L295 Please include in the caption COVID-19 confinement effect or a similar formula, to allow the graphic being self-explanatory
L385. Please, shorten the sentence, or make more stops, it makes 9 lines without a dot. It is difficult to read.
Author Response
Responses to Reviewer II:
Dear authors, I have found your work of interest. However I have a few comments.
Broad comments
- L344 I think that would be of interest discussing directly why in professionals working with animals time spent with their own cats was not increased. Seems a valuable result of the study. The generalist term "special working permissions" seems harder to understand
We thank the reviewer for the comment. At line 355 we specified it better.
- In the regression graphics, I would suggest for clarity including some reference to interpret that increased values comparing with pre-covid values are on the left, and decreased on the right of the estimate 0.0 .
We thank the reviewer for the suggestion. We added a sentence like “Positive estimate values are on the left, while negative estimate values are on the right” to better interpret each of the regression graphs (Line 228, 245, 289, 306)
Specific comments
L165 Please describe the whole words of "AIC", is not so well known as ANOVA or other general terms (e.g)
We wrote the entire word, before the acronym, as suggested
L78 Please describe the first time, the use of QoL, you didn't mention the acronym.
We could not find the QoL acronym at line 78, but we did change the acronym into the entire word at line 115, where we first used it.
L295 Please include in the caption COVID-19 confinement effect or a similar formula, to allow the graphic being self-explanatory
Modified the captions of each graph, as suggested (line 227, 245, 290, 304)
L385. Please, shorten the sentence, or make more stops, it makes 9 lines without a dot. It is difficult to read.
We thank the reviewer for the suggestion. We split the sentence in two. Hopefully, it is easier to read (line 395-399)
Reviewer 3 Report
In this manuscript, Riggioet al. investigated the behavior issues in domestic cats during the COVID-19 confinement. While the pandemic significantly reduced the social activities in humans, the cat-owner interactions increased. As a result, the cat behavior remained stable or even improved during the pandemic. There are other studies of cat-owner relationship during the pandemic, and they are properly cited in this research. Statistical methods were described adequately and were properly performed. This research provides interesting in sights in cat management and welfare.
Please find the specific comments below:
Line 61, add space after [9].
Line 73, add a comma after [8].
Line 107, what proportion of the responses were cat only? What percentage was dog only? What percentage of households has both cat and dog? Do you exclude households with both cat(s) and dog(s)?
Line 165, add a comma at the end of this line.
In the main figures, the authors might want to consider adding the significance in them, although they are already described in the text. I also suggest that the authors include the scatterplot to visualize the spearman’s correction in supplemental figures.
Author Response
RESPONSES TO REVIEWER III
In this manuscript, Riggio et al. investigated the behavior issues in domestic cats during the COVID-19 confinement. While the pandemic significantly reduced the social activities in humans, the cat-owner interactions increased. As a result, the cat behavior remained stable or even improved during the pandemic. There are other studies of cat-owner relationship during the pandemic, and they are properly cited in this research. Statistical methods were described adequately and were properly performed. This research provides interesting insights in cat management and welfare.
We thank the reviewer for their comments and suggestions
Please find the specific comments below:
Line 61, add space after [9].
Done.
Line 73, add a comma after [8].
Correction made as suggested
Line 107, what proportion of the responses were cat only? What percentage was dog only? What percentage of households has both cat and dog? Do you exclude households with both cat(s) and dog(s)?
We have better specified how the survey was built at lines 105-107, and the complete version of the questionnaire has been added as supplementary according to the reviewers’ suggestion, and there question numb. 8 investigates the number of dogs and cats living in the household. Related results are reported at lines 195-196, where it can be seen that in 39.8% of cases, at least one dog was also present (the remaining corresponds to households with only cats)
Information on % of respondents who decided to reply for dogs has been reported at lines 110-112.
Line 165, add a comma at the end of this line.
We are afraid we could not find this typo.
In the main figures, the authors might want to consider adding the significance in them, although they are already described in the text. I also suggest that the authors include the scatterplot to visualize the spearman’s correction in supplemental figures.
We have added an asterisk to indicate significant comparisons and we have indicated the terms of comparisons in the caption.
We have added the scatterplot in the supplementary material.